# Electrical and Electronic Waste Management Problems in Africa: Deficits and Solution Approach

**Gilbert Moyen Massa and Vasiliki-Maria Archodoulaki ***

Institute of Materials Science and Technology, TU Wien, Gumpendorferstrasse 7, Objekt 8, 1060 Vienna, Austria
* Correspondence: vasiliki-maria.archodoulaki@tuwien.ac.at; Tel.: +43-1-58801-30850

**Abstract:** The lack of proper waste management in developing countries results in environmental pollution and human illness. This review presents the available data on the electronic and electrical waste generated and/or transported in Africa. Particular attention is given to waste treatment and the recycling sector, as well as methods for recovering metals from e-waste. The roles and responsibilities of stakeholders and institutions involved in Africa are discussed. Design for Environment guidelines and Sustainable Product Design Concepts are illustrated to find proper strategies for managing e-waste in general, and for Africa in particular. Raising awareness among national and international institutions is necessary to improve e-scraps management in Africa. Measures should be taken to facilitate the transition of e-waste management from the informal to the formal sector, which will create decent jobs and corresponding incomes.

**Keywords:** recycling; Design for Environment (DfE); sustainability metric measurement; e-waste valorization; hazardous substances; Africa; urban mining; Perceived Behavioral Control (PBC); Theory of Planned Behavior (TPB)





## 1. Introduction

Littering has concerned humanity for decades [1]. Organic and biodegradable waste has not always been an urgent concern for humans and the environment. However, as the world population grows, so too does the production of goods to meet demand [2]. Unfortunately, the increased consumption of various goods has led to a substantial increase in electrical and electronic waste, which has become a serious threat to humans, animals, and the ecosystem due to the toxic substances contained within them [3]. The management of municipal solid waste (MSW) is a major problem worldwide, especially in developing countries [4]. Due to a lack of funding, interest in solutions, efficient urban planning, poor equipment for waste collection, and increasing city populations, waste management has become a serious health and environmental issue in developing country municipalities. At the end of their life cycle (EoL), goods need to be either disposed of or appropriately processed with material and/or energy recovery [5]. This step is crucial to avoid negative impacts on the Earth and marine pollution, which can have serious consequences for the environment and people. Admittedly, Africa has the lowest per capita generated e-waste rate in the world. The predominance of the informal sector in many African countries (accounting for more than half of the GDP in many of these countries [6]) has led to a deterioration of the situation in the case of waste of electrical and electronic equipment (WEEE). Design for Environment (DfE) [7] guidelines, along with sustainable product design measurement metrics [8], can be used to find suitable solutions to the problem in Africa. This work presents a solution approach based on the idea of a "circular economy" with the aim of recovering materials contained in products after their use time through proper recycling of e-waste according to the state-of-the-art and adapted to local realities.

## 2. E-Scraps Generated

It is generally known that, in the case of reuse, the owner of a functioning device hands it to a third party by sale or donation (second-hand user) after some service life. The devices are only tested and cosmetically cleaned without further disassembly or replacement of parts. Repair is a necessity linked to a defective product after some service life. In the case of Repair-and-Reuse, the owner of a product no longer covered by a guarantee can either repair it for self-reuse or sell it to a third party that, depending on economic conditions favorable to the buyer, can repair, upgrade, or refurbish it before putting it back on the market [9]. The repair phase is the last step before entry into a recycling process. The difference between them is that by recycling, the original product will completely disappear (be discarded) to obtain some components/materials or/and energy recovery (suitable disposal), while by repair, the device still exists. Refurbishment can be defined as repairing an old product by upgrading it and making it a new product different from the old one. Recycling describes the physical and/or chemical processing of collected product waste with the main aim of recovering materials and/or energy contained in the products at the end of life [10]. The best available technology should be used to quantitatively and qualitatively minimize the residues obtained at the end of the waste treatment process, if this cannot be completely avoided. The life cycle (LC) of a product begins with raw materials extraction, followed by production in factories, the consumer use phase, the waste management phase, and the final waste disposal at the end of life (EoL) of the product [11]. Encouraging both "Reuse" and "Repair-and-reuse" to keep electrical and electronic products alive for as long as possible before bringing them into the recycling process extends the lifespan of products. This is especially important since the consumer use phase, in many cases, is very short.

E-waste [12] covers a large spectrum of valuable electrical and electronic products incorporating non-precious metals (iron, steel, copper, aluminum, etc.), precious metals (gold, silver, palladium, platinum, etc.), plastics, and hazardous substances (e.g., lead-containing glass, mercury, cadmium, batteries, flame retardants, chlorofluorocarbons, and other coolants with the potential to greatly impact the environment). In fact, the outputs of e-waste after treatment generally look, by weight, as follows: 38.1% ferrous metals, 16.5% non-ferrous metals, 26.5% plastic, and 18.9% other [13]. Figure 1 shows the worldwide generated e-waste per region and per waste stream in 2019 (53.6 Mt e-waste was generated in total). It is worth mentioning that since 2014, only the screens and monitors category has decreased (-1%), while the other five stream categories have increased in quantity between 2% and 7%. Secondary raw materials are materials that have been generated from the processing of waste materials to substitute the use of primary materials. Resource recovery from the use of secondary raw materials makes the conservation of primary ores possible, significantly reducing the carbon and ecological footprints. Much of the literature has focused on the LC of daily products like mobile phones, notebooks, desktops, televisions (TVs), and refrigerators/washing machines [14–21]. In the USA, 9% of all aluminum, 21% of beryllium, 19% of copper, 40% of gold, and 26% of silver were used in the EEE manufacturing industry in 2019 [22]. Rare-earth elements (REEs) are commonly used in digital technologies such as disk drives and communication systems, but also in batteries and fuel cells for hydrogen storage, catalysts, light-emitting diodes (LEDs), and fluorescent lighting [23]. In 2018, the recycling rate of REEs was around 1% [24] due to their relatively low prices, but the demand for some REEs surpassed their supply and continues to increase, making their recycling and/or seeking of alternatives an important matter. The concentration of REEs greatly varies depending on the type of e-waste [25]. The unique magnetic and electronic properties of REEs make them abundant in computer hard disk drives, phones, and iPods. Hard disk drives have the highest content of any sample, including neodymium (Nd) > lanthanum (La) > praseodymium (Pr) > dysprosium (Dy) > gadolinium (Gd), with each element ranging from 0.01-0.2% [25]. Erbium (Er) and Thulium (Tm) are the rarest detectable REEs in e-waste samples [25]. Table 1 presents some

products with their average weight and estimated lifespan. After their LC, all these devices become obsolete and are considered electrical and electronic waste.

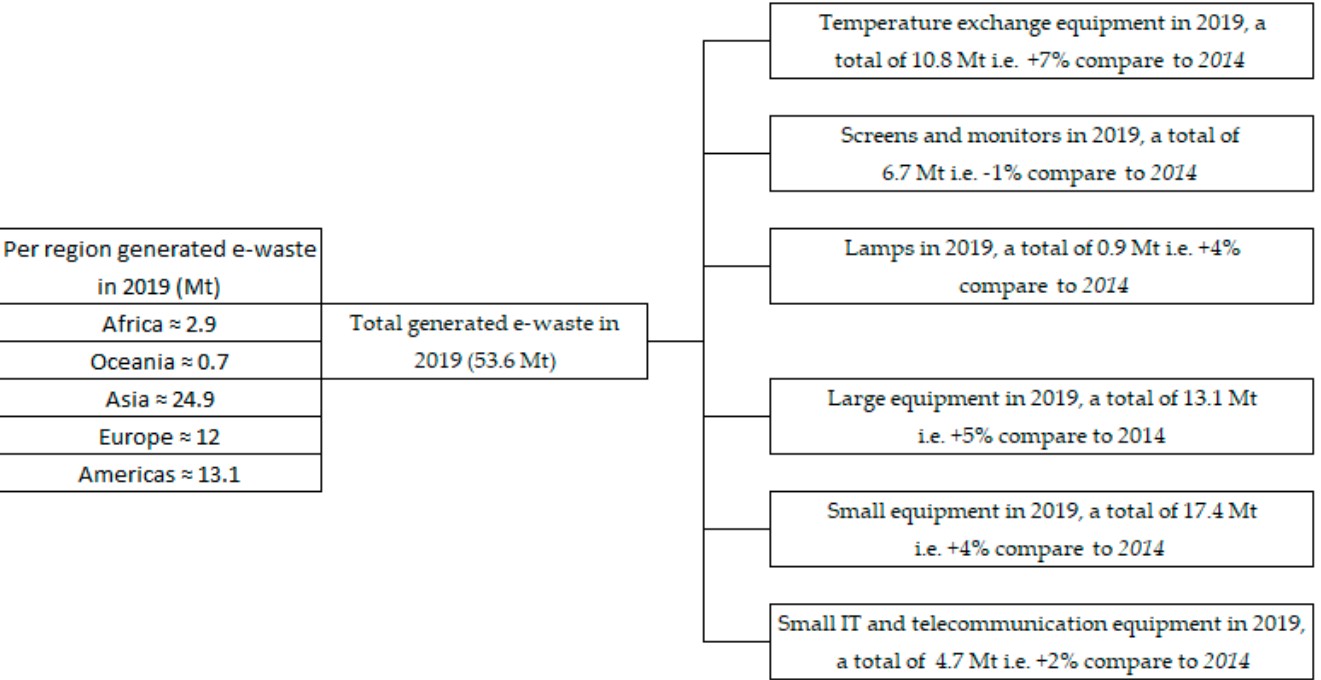

**Figure 1.** E-waste generated worldwide in 2019 by region and waste stream category [26].

**Table 1.** Average weight and estimated lifespan of devices mentioned above [21].

| Item | Average Item Mass (kg) | Estimated Lifespan (years) |
|---|---|---|
| Cell/Mobile phone | 0.1 | 2 |
| Notebook * | 2.3 | 4 |
| Desktop computer | 25 | 5 |
| Television | 30 | 5 |
| Refrigerator | 35 | 10 |
| Battery * | 0.055 | 3.5 |

* Estimation by measurement.

## 3. Special Case of Batteries

Batteries are one of the most important and critical components of electrical and electronic equipment (EEE). Lithium-ion (Li-ion) batteries are one of the most used cells in Europe and, more broadly, the world. Currently, three different Li-ion cell types exist, namely cylindrical, prismatic, and pouch cells [27]. They can be found in various applications, such as mobile/cell phones, laptops, tablets, and in the automotive sector, such as e-mobility, electric vehicles (EV), hybrid electric vehicles (HEV), and plug-in hybrid electric vehicles (PHEV). Additionally, the world of batteries comprises non-rechargeable (also called primary batteries, such as Zinc Carbon "ZnC", Alkaline Manganese "AlMn", Zinc Air "ZnAir", Silver Oxide "AgO", and Lithium Manganese Dioxide "LiMnO$_2$" batteries) and rechargeable (also called secondary batteries, such as Nickel Cadmium "NiCd", Lead-Acid, Nickel Metal Hydride "NiMH", lithium-ion "LiB", and Li-ion-polymer "Li-Po" batteries) [27]. Lead, manganese, nickel, cadmium, lithium, to name a few, can cause health problems. The batteries, which can sometimes be very small, are dispersed and can be found everywhere. Children can encounter these substances and contract diseases [28]. Common electronic items and their components, such as batteries, switches, relays, and printed circuit boards, may contain antimony, barium, beryllium, cadmium, copper, gold,

lead, lithium, mercury, nickel, silver, palladium, and zinc [29–31]. Items are also known to contain a variety of organic chemicals and rare earth metals, the health effects of which have not been studied. Cobalt, nickel, manganese, and lithium are important materials that can be recovered through battery waste recycling. Australia, with 44.8%, and Chile, with 33.3%, produce about 78% of the global lithium supply [27], including electrical and electronic devices, but also hybrid and electric vehicles. Furthermore, 98% of the world's cobalt supply is mined as a byproduct of 61% copper and 37% nickel production, mostly in the Democratic Republic of Congo in Africa [27]. In an environmentally friendly way, Umicore Recycling Solutions operates using a special in-house developed Val Eas process with an annual capacity of more than 4000 tons [32] to treat Ni-metal hydride and Li-ion batteries (battery applications dominate, with 39% of the global lithium markets, followed by ceramic and glass applications).

## 4. E-Waste Valorization and Toxic Substances

In addition to all the hazardous substances present in e-waste, the manufacturing of mobile phones and personal computers consumes significant amounts of gold (Au), silver (Ag), and palladium (Pd) annually mined worldwide. The electronics industry is the third-largest consumer of gold, accounting for 12% of the total gold demand [33]. Table 2 presents a summary of typical pyrometallurgical and hydrometallurgical methods for the recovery of metals from e-waste, as well as some associated toxic substances and diseases. In 2019, 17.4% of e-waste was documented to be collected and recycled, with a potential raw material value of US$10 billion [26]. It was estimated that 4 million tons of secondary raw materials could have been obtained through recycling in 2019. By solely focusing on iron, aluminum, and copper, and comparing emissions resulting from their use as virgin raw materials or secondary raw materials, recycling these materials has helped save 15 million tons of $CO_2$ equivalent emissions in the same year [26]. Photovoltaic modules contain a high percentage by weight of a single element aluminum, while PCBs (Print Circuit Boards) are composed of a mixture of different metals, principally copper, iron, aluminum, tin, and nickel (with an average of 18 elements from the periodic table). Hard disk magnets, while they may contain high amounts of iron, also contain significant amounts of rare-earth elements, particularly neodymium, praseodymium, and dysprosium. For example, since photovoltaic modules contain less than 1% [22] of silver in their composition, it can become economically profitable to recycle them. According to the "Global Alliance for Incinerator Alternatives (GAIA)" [34], after incineration, about 30% of air pollutants still remain deposited in landfill as fly ash, bottom ash, boiler ash, slag, and wastewater treatment sludge, affecting future generations.

**Table 2.** Typical hydrometallurgical/pyrometallurgical processes for recovery of valuable metals from e-waste and associated toxic substances [35–45].

| Industrial Processes | Metals Recovered | Main process Features | Main Metallurgical Process | Toxic Substances | Exposure Route | Average Estimated Concentration in e-Waste (mg/kg) * | Health Effects (a Few Diseases) |
|---|---|---|---|---|---|---|---|
| Noranda process at Quebec, Canada | Cu, Au, Ag, Pt, Pd, Se, Te, and Ni | Smelting of e-waste and Cu concentrate (14% of the total throughput). Electrorefining for metal recovery | Pyrometallurgy | **Persistent organic contaminants** | | | |
| | | | | Brominated flame retardants | Air, dust, food, water, and soil | | Thyroid problem, impaired development of the nervous system etc. |
| Boliden Rönnkär Smelter | Cu, Ag, Au, Pd, Ni, Se, Zn, and Pb | Smelting in Kaldo reactor, upgrading in Cu and high Precious Metals recovery by copper refining. Total feed 100,000 tons every year | Pyrometallurgy | Polybrominated diphenyl ethers (PBDEs) | | | Reproductive neurobehavioral development, thyroid function. Hormonal |
| | | | | **Polybrominated biphenyl (PBBs)** | | | |
| Test at Boliden Rönnkär Smelter | Copper and precious metals (PMs) | PC scrap feeding to a zinc Fuming process (1:1 mixture with crushed revert slag); Plastics were tested as reducing agent and fuel; Copper and precious metals following the cop per collector to be recovered to the copper smelter | Pyrometallurgy | Polychlorinated biphenyl (PCBs) | Air, dust, food, and soil (bio-accumulative in fish and seafood) | | Carcinogenicity, on multiple targets such as liver, thyroid, immune function, repro duction, and neurobehavioral development. |
| Umicore´s Precious metal refinery at Hoboken, Belgium | Au, Ag, Pd, Pt, Se, Ir, Ru, Rh, Cu, Ni, Pb, In, Bi, Sn, and Sb As, | IsaSmelt, copper leaching, and electrowinning and precious metal refining for Precious Metal Operation (PMO); E-waste cover up to 10% of the feed (250,000 tons of different wastes per annum); Plastic partially substitutes the coke as reducing agent and fuel in IsaSmelt; existence of Offgas emission control system | Combination of pyrometallurgy and hydrometallurgy | **Dioxins** | | | |
| | | | | Polychlorinated dibenzodioxins (PCDDs) and dibenzofurans (PCDFs) | Air, dust, food, water, soil, and vapour | | Reproductive, neurobehavioral and immune development |
| | | | | Polyaromatic hydrocarbons (PAHs) | Released as combustion byproduct: air, dust, soil, and food (bio accumulative in fish and seafood) | | Carcinogenicity, mutagenicity, and teratogenicity |

**Table 2.** *Cont.*

| Industrial Processes | Metals Recovered | Main process Features | Main Metallurgical Process | Toxic Substances | Exposure Route | Average Estimated Concentration in e-Waste (mg/kg) * | Health Effects (a Few Diseases) |
|---|---|---|---|---|---|---|---|
| | | | | **Heavy metals** | | | |
| Umicore's trial | Au, Ag, Pd, Pt, Se, Ir, Ru, Rh, Cu, Ni, Pb, In, Bi, Sn, As, and Sb | Plastics-rich material from WEEE were tested to replace coke as a reducing agent and energy source for the IsaSmelet | Combination of pyrometallurgy and hydrometallurgy | Lead (Pb) | Air, dust, food, water, and soil | 1782.4 | Neurobehavioral development of children. Anemia. Kidney damage. Chronic neurotoxicity |
| | | | | Chromium (Cr) or hexavalent chromium | Air, dust, food, water, and soil | 75.5 | Carcinogenicity, Reproductive functions. |
| | | | | Cadmium (Cd) | Air, dust, food, water, and soil (specially rice and vegetables) | 39 | Endocrine function. Ovotoxicity |
| Dunn's patent for gold refining | Au | Gold scrap reacted with chlorine at 300 °C to 700 °C; Hydrochloric acid to dissolve the impurity-metal chlorides; Ammonium hydroxide and nitric acid washing respectively to dissolve the silver chloride; Samples should contain more than 80% of gold | Combination of pyrometallurgy and hydrometallurgy | Mercury (Hg) | Air, dust, food, water, and soil (bio accumulative in fish) | 1.2 | Neurobehavioral development of children (especially methylmercury). Anemia. Kidney damage |
| Outotec's Ausmelt TSL and Kaldo Furnaces | Zn, Cu, Au, Ag, In, Pb, Cd, and Ge | Copper scrap and e-waste recycling with many refining steps downstream | Pyrometallurgy | Zinc | Air, dust, food, water, and soil | 1561.1 | Increased risk of Cu deficiency (Anemia, neurological abnormalities) |
| | | | | Nickel (Ni) | Air, water, soil, and food (plants) | 65.8 | Carcinogenic, lung embolism, respiratory failure |
| | | | | Lithium (Li) | Air, water, soil, and food (plants) | 44.3 | Burning sensation, Cough. Laboured breathing |
| | | | | Barium (Ba) | Air, dust, and water | 626 | Increased blood pressure, stomach irritation, nerve |

**Table 2.** *Cont.*

| Industrial Processes | Metals Recovered | Main process Features | Main Metallurgical Process | Toxic Substances | Exposure Route | Average Estimated Concentration in e-Waste (mg/kg) * | Health Effects (a Few Diseases) |
|---|---|---|---|---|---|---|---|
| Dowa mining Kosaka Japan | Cu, Au, and Ag | E-waste TSL, smelting in a secondary copper process | Hydrometallurgy | Beryllium (Be) | Air, water, and food | 0.014 | Pneumonia. Berylliosis a persistent and lung problem |
| | | | | Aluminum (Al) | Air, dust, water, and soil | 9510 | Skeletal development and metabolism, neurotoxicity, fetal toxicity |
| | | | | Antimony (Sb) | Air, water, and soil | 180 | Damage lung, heart, liver, and kidney, eye irritation, etc. |
| | | | | Arsenic (As) | Air, soil, water, and food | 0.47 | Skin alterations. Decreased nerve, diabetes, cancer |
| | | | | Bismuth (Bi) | Air, water, and soil | 2.7 | Kidney damage, serious ulceration stomatitis, albumin, etc. |
| I.S-Nikko's recycling facility, Korea | Au, Ag, and Platinum Group Metals | Recycling in TSL, smelting followed by electrolytic refining | Pyrometallurgy | Cobalt (Co) | Air, dust, water, soil, and food | 8.3 | Discomfort of bodies, albumin, diarrhea, etc. |
| | | | | Copper (Cu) | Air, dust, water, and soil | 115.5 | Asthma, pneumonia, nausea, vision and heart problem, etc. |
| | | | | Gallium (Ga) | Air, water, and fume | 2.43 | Irritation of the nose, mouth, and eyes, headache, diarrhea |
| Day's Patent | Pt, Pd, and Precious Metals | Smelting in plasma arc furnace at 1400 °C. PMs collected in Basis Metal (BM). Ag and Cu used to collect metal | Combination of pyrometallurgy and hydrometallurgy | Germanium (Ge) | Air and dust | 1.9 | Abdominal cramps, burning sensation, red skin and eyes |
| | | | | Indium (In) | Air, dust, water, and soil | 4.6 | Damage the heart, kidney and liver, etc. |
| | | | | Molybdenum (Mo) | Air, dust, water, and soil | 1.2 | Liver disfunction with hyperbilirubinemia, pain in knees, etc. |
| | | | | Selenium (Se) | Air, dust, water, and soil | 12.67 | Hair loss, cardiovascular, renal, and neurological problem |

**Table 2.** *Cont.*

| Industrial Processes | Metals Recovered | Main process Features | Main Metallurgical Process | Toxic Substances | Exposure Route | Average Estimated Concentration in e-Waste (mg/kg) * | Health Effects (a Few Diseases) |
|---|---|---|---|---|---|---|---|
| Aleksandrovich Patent | Au and Platinum Group Metals | Scrap combustion in a BM with carbon reduction | Pyrometallurgy | Silver (Ag) | Water and soil | 49 | Allergic dermatitis, inhalation hazards |
| | | | | Tin (Sn) | Air, dust, water, and soil | 1716.4 | Eye and skin irritation, headache, stomach ache, etc. |
| | | | | Vanadium (V) | Air, dust, water, and soil | 66 | Severe eye, nose, and throat irritation |
| Aurubis recycling Germany | Cu, Pd, Zn, Sn, and Precious Metals | Smelting of Cu and e-waste in TLS, black Cu processing and electrorefining | Hydrometallurgy | Yttrium (Y) | Air, dust, water, and soil | 1.99 | Lung embolisms, cancer with humans |
| | | | | Iron (Fe) | Air, dust, water, and soil | 91.1 | Liver damage |

* Values from different sources and areas.

## 5. E-Waste Recycling Process

After size reduction/comminution, the following types of separation technologies can be implemented [13]: corona-electrostatic and eddy-current separation, based on the difference in the electrical conductivity of the materials; magnetic separation, consisting of separating metals based on their magnetic properties; gravity separation (also called density-based separation), which depends on the density and particle size; and optical separation, all with the aim of refining and detoxifying the various outputs of the pre-processing. The following metallurgical processes for recycling exist: hydrometallurgical, pyrometallurgical, and bio-metallurgical processes, as well as combinations of these. They can all be used to process the output of preprocessing WEEE. The first two processes [13,46] are currently the major routes for e-waste processing with materials recovery, and there are only a few laboratory studies for e-waste treatment through bio-metallurgical processes. However, bioleaching of metals from e-waste has the potential for further improvement. Hydrometallurgical recovery processes of metals involve oxidative leaching for metals extraction, followed by separation and purification. Its advantages over thermal treatment/pyrometallurgy include lower toxic residues, lower emissions, and higher energy efficiency. Hydrometallurgical processes are based on traditional hydrometallurgical technology for metals extraction from primary ores [46]. Due to its cost-effectiveness and environmental efficiency, biotechnology [12] will play a significant role in the future of e-waste treatment and material recovery.

To improve material recovery rates without negatively impacting the environment, more investment in advanced technologies, especially in metal recovery, is required for the state-of-the-art end-processing of e-waste. However, this is not currently a realistic solution for developing countries like many African countries that lack the financial resources or management necessary for development. For example [12], a typical aluminum smelter in Europe requires a minimum input of 50 thousand tons of aluminum scrap per year and an investment cost of about €25 million to run a plant. Only a few companies in the world, such as Aurubis AG in Germany, Boliden in Sweden, DOWA in Japan, Umicore in Belgium, and Xstrata in Canada, are equipped with the technical know-how, sophisticated flow sheets, and sufficient economy of scale for precious metal refinery to fulfill technical and environmental requirements. The integrated smelter-refinery of Umicore Precious Metal Refining in Belgium has the capacity to produce 2400 tons of silver, 100 tons of gold, 25 tons of palladium, and 25 tons of platinum per year at an investment cost of more than €500 million [12]. About 25% of the annual production of silver (Ag) and gold (Au), and 65% of Palladium (Pd) and Platinum (Pt), come from e-waste and end-of-life catalysts [33]. In addition, the recovery of metals from electrical and electronic equipment mitigates the high $CO_2$ emissions associated with primary metal production. The $CO_2$ emissions of the Umicore process [32], when recovering 75,000 tons of metal from 300,000 tons of valuable materials and smelting byproducts, are only 3.73 tons of $CO_2$/ton of metal compared to 17.1 tons of $CO_2$/ton of metal using a primary production route. The continuous improvement of these measures leads to very low emissions and prevents the loss of precious metal dust. Recycling of e-waste needs to be encouraged worldwide because of the significant energy savings from using recycled materials compared to using virgin materials, as presented in Table 3.

**Table 3.** Energy saved by using recycled materials over virgin materials [47].

| Material | Energy Savings (%) |
|----------|--------------------|
| Aluminum | 95 |
| Copper | 85 |
| Iron and steel | 74 |
| Lead | 65 |
| Zinc | 60 |

**Table 3.** *Cont.*

| Material | Energy Savings (%) |
|:---:|:---:|
| Paper | 64 |
| Plastic | >80 |

## 6. Strategies for Africa

The African continent, with its 54 countries [48], is one of the largest (30.37 million km$^2$ [48]) and most populous (1.4 billion [49] inhabitants estimated in 2021) continents on Earth. Africa [48] represents around 6% of the Earth's total surface area, 20% of its land area, and 18% of the global population. The average annual population growth rate is more than 2% [50], and the average population density is 46.1 inhabitants per km$^2$ [48]. The illegal trade in waste electrical and electronic equipment is also a significant worldwide transcontinental concern. Ghana and Nigeria in Africa are among the biggest recipients of e-waste from developed countries. It is estimated that around 500 containers [51] of electrical and electronic equipment enter Nigeria every month. According to the same source, approximately 400,000 used computers are imported every month, of which only around 50% still function (45% of the equipment comes from Europe, 55% from the US, and 10% from Asia). The same source mentions that approximately 300 containers of used and/or waste electrical and electronic equipment arrive at the ports of Tema in Ghana every month, and that on average, 75–80% of the imported used and/or waste electrical and electronic equipment cannot be reused. South Africa, which is one of the emerging African economies in the world and a member of the BRICS (Brazil, Russia, India, China, and South Africa) group, is facing a significant e-waste problem (5.4 kg/per inhabitant) [52] in terms of massive generation and inadequate management mechanisms, with enormous environmental challenges. In Ghana (Agbogbloshie) and Nigeria (Alaba), crude methods such as burning are used to retrieve precious metals and reusable components [51]. There is no formal legislation to manage and enforce WEEE management in Egypt. Electronic waste is mainly dealt with by the informal sector, and after extracting the recyclable streams, it is generally either burned or thrown into landfills/dump sites in slums such as Manshiet Nasser [53]. Rwanda is a country in East Africa with well-structured e-waste management in Africa. Rwanda has a law based on licenses (license 1 for collection and transportation service, license 2 for dismantling and refurbishment service, and license 3 for recycling service) for any person or group of persons who wants to do business in this domain. There are also considerable fines for those who do not respect the legislation [54]. In 2021, the African population was estimated to be 1.4 billion, a number that is predicted to grow to approximately 1.7 billion by 2030, associated with a population growth of 21.4% from 2021 to 2030. In 2021, approximately 54.8 million tons (an average of 52.2 and 57.4 tons) [26,33,55] of e-waste was supposed to be generated worldwide. The worldwide prediction for 2030 according to "The Global E-waste Monitor 2020" [26] is 74.7 million tons, which means an increase in the quantity of e-waste of 36.3%. Assuming that the ratio of the amount of worldwide generated e-waste over the amount of generated e-waste in Africa in 2021 remains equal to the ratio of the predicted values in 2030, then the estimated amount of generated e-waste in Africa for 2021 is 3 million tons, increasing to 4 million tons in 2030, as shown in Table 4.

**Table 4.** In 2019, the world generated millions of tons of e-waste. Comparative values for Africa are reported as estimated values for 2021 und predicted values for 2030 [26,33,55].

| | 2019 | 2021 | 2030 |
|:---:|:---:|:---:|:---:|
| Worldwide | 53.6 | 54.8 | 74.7 |
| Africa | 2.9 | 3 | 4 |

The predicted 4 million tons for 2030 have to be considered with a large estimation error (underestimation); nevertheless, it still corresponds to an e-waste growth rate of 33.3%

for the considered period (2021–2030). This means that e-waste generation is growing at least 1.5 times faster than the African population over the same period, despite low accuracy estimations for electronic waste in 2030. In Africa, an average of 2.5 kg [26] of e-waste per capita and a total of 2.9 Mt of e-waste were generated in 2019. According to Table 4, around 3 Mt of e-waste was generated in Africa in 2021. If we assume that the quotient of total generated e-waste and the amount of per capita generated e-waste in 2019 in Africa is slightly equal to the same quotient in 2021, then per capita generated e-waste in 2021 will be equal to 2.6 kg. Figure 2 shows the generated e-waste in some countries in each African subregion and per inhabitant in the same year.

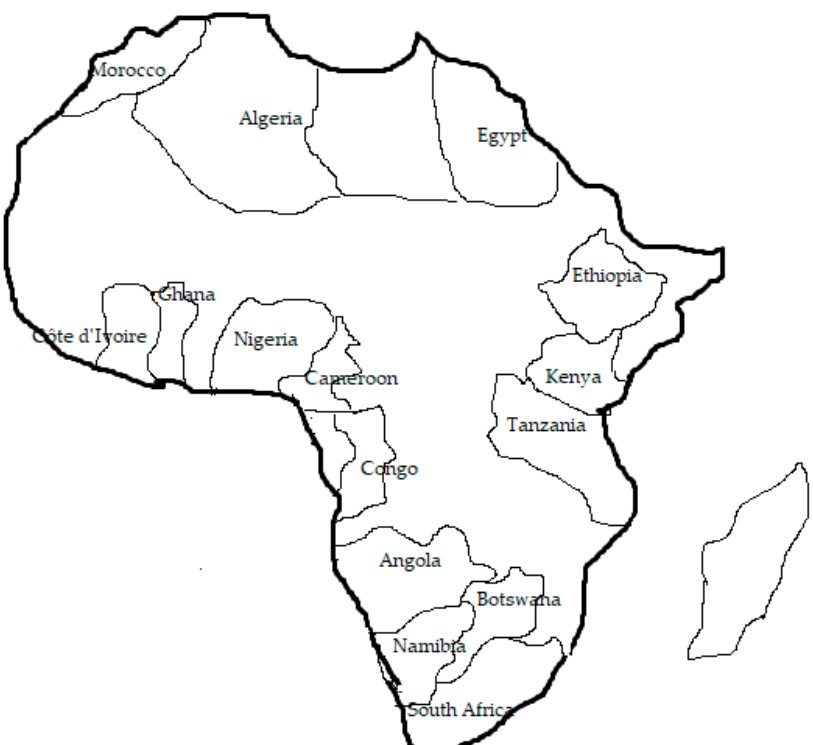

Eastern Africa: 0.3 Mt generated and 0.8 kt/capita
Ethiopia 55.2 kt
Kenya 51.3 kt
Tanzania 50.2 kt

Middle/central Africa: 0.2 Mt generated and 2.5 kt/capita
Angola 125.1 kt
Cameroon 26.4 kt
Congo 18.3 kt

Northern Africa: 1.3 Mt generated and 5.4 kt/capita
Egypt 585.8 kt
Algeria 308.6 kt
Morocco 164.5 kt

Southern Africa: 0.5 Mt generated and 6.9 kt/capita
South Africa 415.5 kt
Botswana 18.8 kt
Namibia 15.7 kt

Western Africa: 0.6 Mt generated and 1.7 kt/capita
Nigeria 461.3 kt
Ghana 52.9 kt
Côte d'Ivoire 30.0 kt

**Figure 2.** E-waste generated in 2019 by sub-region/inhabitant in Africa [26].

The implication of e-waste management for a country or region is the need to establish well-organized logistics and a database to address the rising e-waste in the area. There are currently two useful e-waste collection systems in developed countries [56]: (1) a collective system, usually founded as a nonprofit and nongovernmental organization by trade associations, which focuses on some product categories to efficiently find a market for their reuse; and (2) a clearing house system where producers, recyclers, waste businesses, and others compete to provide services. There are three commonly used channels for e-waste logistics [56]: (1) municipal collection sites, where citizens can deposit any amount of waste at no cost; (2) in-store retailer take-back schemes, which may be free or depend on repeat purchases; and (3) direct producer take-back, which is usually for business customers and may require a replacement purchase.

Currently, data on e-waste recycling companies in Africa are old and rare, which can be explained by the lack of transparency of actors in the sector and a sign that the sector is still informal in many countries. In Africa, it is documented that only 0.9% of the 2.9 Mt of generated e-waste [26] was collected in 2019, and it was estimated [26] that, in the same year, 55.2 kt e-waste was generated in Ethiopia, 51.3 kt in Kenya, 50.2 kt in Tanzania (in the east), 125.1 kt in Angola, 26.4 kt in Cameroon, 18.3 kt in Congo (in central), 585.8 kt in Egypt, 308.6 kt in Algeria, 164.5 kt in Morocco (in the north), 415.5 kt in South Africa, 18.8

kt in Botswana, 15.7 kt in Namibia (in the south), 461.3 kt in Nigeria, 52.9 kt in Ghana, and 30.0 kt in Côte d'Ivoire (in the west), among other relevant countries.

In 2017, Nigeria generated about 288,000 tons of e-waste [57], and in Ghana, about 15% of the imported electrical and electronic devices in 2009 were not functioning. Four companies were found to be mainly involved in metal recycling, namely Atlantic Recycling (which operates on repair and re-use activities), City Waste Recycling, FIDEV Recycling (which operates on dismantling and trading of scrap metals), and Blancomet Recycling (which operates on dismantling and trading of scrap metals). However, there is no information concerning the quantity of treated waste [57].

In 2015, approximately 17,733 tons of WEEE were collected and recycled across 27 recycling companies in South Africa [57]. Of these companies, 79% were comprised of ICT and consumer electronics. In 2018, 45.6 million mobile subscribers were identified in Kenya, and recycling was carried out from both dumpsites and primary collection sites [57]. In 2016, 97.8 million mobile subscribers were identified in Egypt [57]. The international Technology Group, Recycle Bekia, and Eco Integrated Industrial Systems can be seen as emerging companies in recycling here, but the informal sector is dominant. In Africa, all these companies are active in recycling for the winning of metal.

Particularly, the transboundary movement of old devices from developed to developing countries needs to be addressed. It is estimated that 16 to 38% of WEEE collected in the EU and 80% in the U.S. are sent "legally and/or illegally" to developing countries in the form of reused or discarded devices [13]. At least one-third of the 2.2 Mt [33] of African e-waste quantity on average was estimated to have been illegally imported in 2016. It is necessary to fight dispersion, contamination, and the loss of target materials to undesirable streams [13]. Manual disassembly provides the best recovery rate of original components and materials without damaging them, making it easier to sort and improve their reuse. To achieve this, we need to follow the 76 Design for Environment (DfE) guidelines defined by Telenko C. et al. [7], and select those that ensure that all EEE sold in the African market fulfill the desired design. Then, use the metric for Sustainable Product Design Concepts measuring of Han J. et al. [8] to confirm the sustainability of the choice, specifically measuring the material, production, use, and end of life. The mobile phone is chosen for the calculation because the African mobile phone market has shown resilience to the COVID-19 pandemic, it initially declined in the first quarter of 2020 but remained stable in the second, with delivery of 20.1 million smartphones in both quarters. The third quarter showed a resurgence of activity with an increase of 2.8 million units, and in the first quarter of 2021, there were 23.4 million smartphones shipped [58]. In 2017, Nigeria welcomed the continent's first smartphone assembly unit. AfriOne, located in the free zone, produces 120,000 units per month marketed between $92 and $108 to middle-income class consumers, a large part of the tens of millions of consumers in Nigeria [59]. The majority of smartphones sold on the African continent come from abroad, and consequently, the quantity of obsolete mobile phones will continuously increase in the future. Han J. et al. determined the actual value of the negative environmental impacts caused at the conceptual design stage, with a result that could reflect the level of sustainability in a simple but effective manner using the measurement scales low (0), medium (1), and high (2) to indicate sustainability attributes. All parameters used in the equations are defined in Table 5. The authors clearly defined in their work under which conditions each metric category (material, production, use, and end of life) could be calculated to aid decision-making. Based on Design for Environment (DfE) guidelines and sustainable product design measurement metrics, the international community can define criteria that a product must satisfy before entering the international market, thereby solving the environmental problem related to WEEE, or at least substantially reducing it. As some quantities of this e-waste end up in Africa,

the continent can apply those criteria to protect their market and the environment. The $Metric_{Material}$, $Metric_{Production}$, $Metric_{Use}$, and $Metric_{EOL}$ are given by the following equations:

$$Metric_{Material} = \frac{9 \times \left( \frac{\sum_{i=1}^{N}(M_1+M_2) \times M_3}{N} \right)}{8} + 1 \tag{1}$$

$$Metric_{Production} = \frac{9 \times (P_1 \times P_2 + P_3) \times P_4}{12} + 1 \tag{2}$$

$$Metric_{Use} = \frac{9 \times U_1 \times (U_2 + U_3)}{8} + 1 \tag{3}$$

$$Metric_{EOL} = \frac{9 \times (E_1 + E_2 + E_3) \times E_4}{12} + 1 \tag{4}$$

**Table 5.** Evaluation of each metric measurement relating to mobile phones.

| Metrics | Attributes | Business as Usual Production | Under Selected DfE Conditions Production |
|---|---|---|---|
| Material | Material origin ($M_1$) | • Stainless steel (1), screen (1), plastic (1), battery (0), ceramic as composite (0) | • Only recy. stainless steel (2), only recy. * LCD screen (2), recy. PC-plastic (2), recy. battery (1), natural ceramic (1) |
| | Material property ($M_2$) | • Stainless steel (1), screen (1), plastic (1), battery (0), ceramic as composite (0) | • Only recy. stainless steel (2), only recy. LCD screen (2), recy. PC-plastic (2), recy. battery (1), natural ceramic (1) |
| | Use material quantity ($M_3$) | • Stainless steel (1), screen (1), plastic (1), battery (0), ceramic as composite (0) | • Only recy. stainless steel (2), only recy. LCD screen (2), recy. PC-plastic (2), recy. battery (1), natural ceramic (1) |
| | Use of material type (N) | 5 | 5 |
| | $Metric_{Material}$ | 2.4 | 7.3 |
| Production | Balance between the number of parts and complexity ($P_1$) | Currently design standard with a mass production (2) | The same design standard with a few more steps like the production of recycled components (1) |
| | Parts standardisation ($P_2$) | Battery and some components can benefit from standard component (2) | Some components require customisation (0) |
| | Parts design for assembly ($P_3$) | Good potential for assembly (2) | Good potential for assembly (2) |
| | Suitable fabrication method ($P_4$) | Currently valid operations are needed (2) | Relative more operations are needed (1) |
| | $Metric_{Production}$ | 10 | 2.5 |
| Use | Product use time/lifetime ($U_1$) | The design time needs to be closer to its use time (1) | The design time needs to be closer to its use time (2) |
| | Energy consumption during use ($U_2$) | Needs battery to power (1) | Needs also recy. battery to power (2) |
| | Robustness, reliability, and maintenance ($U_3$) | Internal components for the base will require a fair amount of resource to maintain/service (1) | Internal components for the base will require a fair amount of resource to maintain/service (1) |
| | $Metric_{Use}$ | 3.3 | 7.8 |

**Table 5.** *Cont.*

| Metrics | Attributes | Business as Usual Production | Under Selected DfE Conditions Production |
|---|---|---|---|
| | Reuse (E$_1$) | Battery and some components have fair potential to be reused (1) | Battery and some components have great potential to be reused (2) |
| | Recycling, remanufacturing, and repair (E$_2$) | All material involved can be recycled or not (1) | Almost all material involved can be recycled (2) |
| | Disposal (E$_3$) | Blender base that contains battery and some components will not be easy to disassemble (1) | Battery and some components cause a very slight negative impact due to disposal (1) |
| | Ease of disassembly (E$_4$) | Blender base that contains battery and some components will not be easy to disassemble (1) | Blender base that contains battery and some components will be easier to disassemble/landfill (2) |
| End of life | Metric$_{EOL}$ | 2.5 | 8.5 |

* recyclable.

To produce a readily understandable outcome, the authors conducted a scaling process, ensuring that the final value of each metric falls between 1 and 10, where 1 signifies poor and 10 signifies excellent sustainability [8]. The evaluation for mobile phones is presented in Table 5. An overview of each metric category's sustainability for mobile phones is provided in Figure 3.

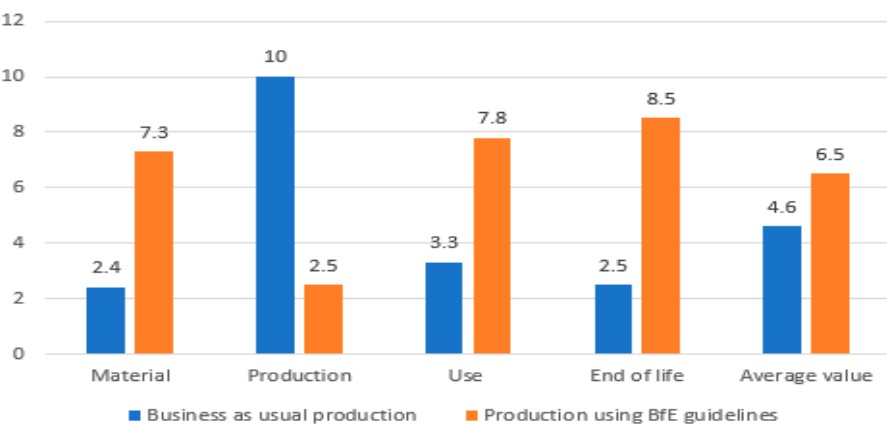

**Figure 3.** Overview of each metric category with respect to sustainability for a mobile phone.

Figure 3 shows that production, based on current practices, receives the highest score of 10. This is simply because we assume that firms find the current production process satisfactory (i.e., profitable) and that the alternative eco-design (DfE) has a low score of 2.5. However, we do not agree with this assessment since, for the alternative approach, all other scores (material, use, and end of life) are greater than 7 and even exceed the average value, resulting in a good environmental rating. Although the financial cost of this alternative has not been evaluated, we believe that given the threat of climate change and its disastrous consequences for humans and the environment, such efforts are worthwhile. This method can be applied to other electronic and electrical equipment (EEE) devices to help designers make environmentally sound decisions at the design stage, considering recycling in material selection to minimize the negative impact of obsolete products on the environment and living beings. Measures and strategies should also be developed to deal with existing electronic scrap. Additionally, Figure 4 presents another way to organize and manage WEEE in Africa by improving current practices. Many EEE companies are beginning to prioritize environmental protection in relation to the products they bring to market, as evidenced by their websites. For example, HP publishes a recycling

vendor list to promote transparency and progress in raising social and environmental standards in the electronics industry supply chain. HP also publishes recycling volumes for their products in various countries and provides take-back services for a broad scope of products [60]. Lenovo offers Asset Recovery Services (ARS) to business customers to manage their IT assets and data center infrastructure, including equipment take-back, data destruction, refurbishment, and recycling services [61]. Dell has recovered over 2.5 billion pounds (1.1 billion kg) of used electronic equipment since 2007, as they encourage people to bring back their old products [62]. Samsung aims to achieve net-zero $CO_2$ emissions, use 100% renewable energy, develop environmentally friendly technologies, conserve and reuse resources, save water, and treat pollutants by 2030 [63]. Huawei is committed to minimizing its environmental impact through recycling and reuse to conserve resources and prevent waste. Electronic waste is recovered by dissolving raw materials such as copper, iron, aluminum, cobalt, etc., to introduce them into the recycling process [64]. However, groundbreaking technologies and marketing strategies are not apparent when looking at all EEE company homepages.

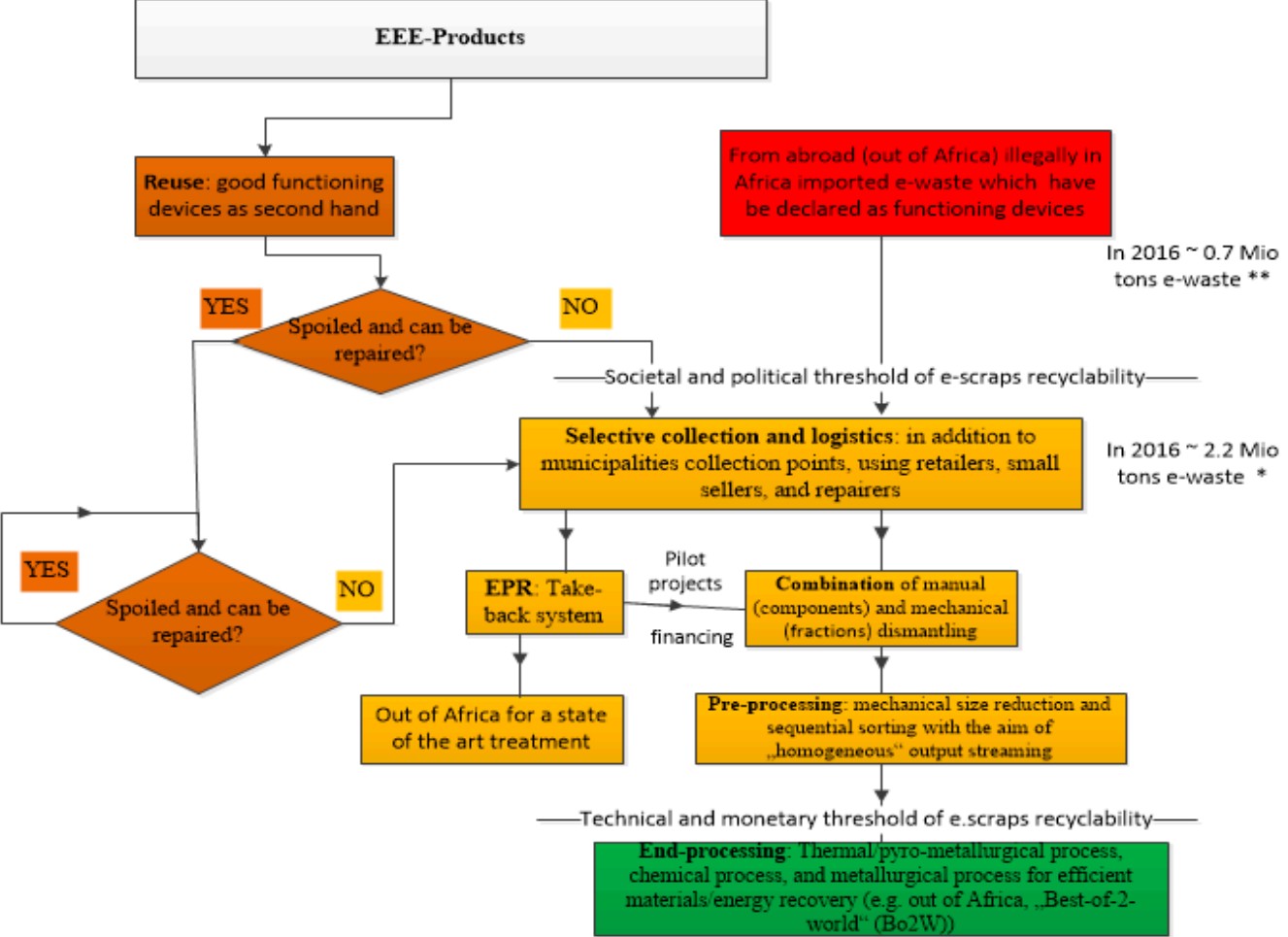

**Figure 4.** E-waste management strategy for Africa (improvement on current practice). Source for * [33] and for ** we assume that around 1/3 of the 2.2 Mt e-waste in Africa in 2016 was illegal importation.

Table 6 below presents some appropriate measures, which need to be put in place now, in short-, middle-, and long-term to solve the problem of e-waste in Africa.

**Table 6.** Current, short-, middle-, and long-term solutions of e-waste problem.

| Appropriate Measures to Be Put in Place |
|---|
| Based on current practice in African countries, a significant portion of e-waste is illegally imported, causing significant harm to the population and the environment. In the short term, this activity needs to be effectively and completely banned, monitored, and sanctioned by national and international law. |
| Short-term strategy (within 5 years) should focus on improving the current situation by implementing a comprehensive and efficient collection and logistics strategy that involves all stakeholders. This should be accompanied by monitoring and raising awareness among all actors involved in the process to ensure proper handling of the waste and promote health and environmental safety. Additionally, motivated by the need to minimize the environmental impact of e-waste, several technological changes have been made. These include: <br> • The replacement of CRT screens with LCD screens (eliminating Pb but introducing Hg) <br> • The introduction of optical fibres (Cu eliminated from the cabling, but F, Pb, Y and Zr introduced) <br> • The introduction of rechargeable batteries (Ni, Cd reduced, but Li increased), and so on. <br> All this changes and their consequences need to be considered during the improvement. A well-organized and structured manual disassembly process for products that are not taken back will also be a part of a sustainable African e-scrap recycling strategy. |
| Meddle-term strategy (from 6 to 30 years) involves gradually organizing e-waste preprocessing up to recycling. This includes reducing landfill, organizing waste handling and utilization services by waste companies country-wide, improving hazardous waste collection, ensuring that hazardous waste packaging and labeling comply with special legislation, transporting hazardous waste only to landfills that can treat them, treating specific types of hazardous waste in Africa, stabilizing waste quantities using charges/taxes, and further reducing waste. An effective take-back system (EPR) or a combination of manual and mechanical disassembly, mechanical size reduction, and sequential sorting systems should be used to obtain homogeneous output streams at the end of the process. With photovoltaic technology being part of the solution for renewable energy, its recycling will become a challenge in 15 years due to the large amount of obsolete solar panels. Pilot projects in cooperation with producers (EPR) and the Climate Change Action Plan (2021–2025) from the World Bank Group (WBG) should be implemented [65]. Financial possibilities should be utilized to set up a policy and transitional legislation that considers e-waste management problems for sustainable development in Africa. The educational system should be reformed starting with a proper program on waste management, and encourage reduce, reuse, repair, and recycling to increase the lifespan of products and save resources. |
| The long-term strategy (from 31 to 50 years) consists of end processing, which is a technical and economic challenge in e-waste treatment. Various processes, such as thermal pyro metallurgical, chemical, and metallurgical, are used for efficient materials and/or energy recovery. Umicore Precious Metal Refining in Belgium has the capacity to produce 2400 tons of silver, 100 tons of gold, 25 tons of palladium, and 25 tons of platinum per year, and the investment cost for the metallurgical processes is more than €500 million. Technical know-how and large investments are necessary to achieve this step, and many individual African countries do not have the capacity to do it alone. The "Best of Two Worlds (Bo2W)" philosophy can be a solution approach for African countries, or many countries can come together and construct the plant corresponding to their needs. "Best of two worlds (Bo2W)" philosophy [12] suggests a pragmatic network solution for e-waste management in emerging economies, which seek technical and logistical integration of "best" manual e-scrap disassembly based preprocessing in developing countries and "best" end processing treatment of hazardous and complex fractions in dedicated facilities in developed countries. Existing technologies should be used to recycle the minimized waste, which occurs when EEE products reach the end of their life cycle. The goal is to gradually and significantly reduce this waste by improving its landfill and take the treatment of electrical and electronic waste in Africa out of its embryonic state. |

## 7. Discussion

To highlight the disparity in e-waste management between developed and developing countries, we conducted interviews and surveys with stakeholders in Vienna (Austria) and Douala/Yaoundé (Cameroon), in addition to reviewing relevant literature. The summarized results of the survey and research can be found in Table 7.

**Table 7.** Overview of the survey and the research [26,66–68].

| | Austria | Cameroon |
|---|---|---|
| Population (in million) | 8.9 | 26.6 |
| Considered big cities and its popuulation (in milion) | Vienna ≈ 1.9 | Douala (Dla) ≈ 3.5 <br> Yaounde (Yde) ≈ 4.1 |
| Municipal Solid Waste (MSW) quantity per year | Vienna ≈ 1,024, 407 tons, 549 kg/capita | Dla ≈ 694,483 tons, <br> Yde ≈ 2/3 of Dla quantity, 226.3 kg/capita for both cities |

**Table 7.** *Cont.*

|  | **Austria** | **Cameroon** |
| --- | --- | --- |
| Considered plastic and EEE waste quantity per year in those cities | Plastic ≈ 8195 tons<br>EEE ≈ 8333 tons (from DRZ-Vienna) | Plastic ≈ 20,884 (3% of 694,483) tons for Dla and 13,890 tons for Yde<br>EEE ≈ 2/3 of 26.4 kt for both cities |
| Existence of well organized waste selection and collection | Yes | No |
| EEE devices disassembly time | 1–3 h depending on devices | Bad disassembly activity, dangerous burning to gain copper for example |
| EEE repair time of devices | 1–3 h depending on devices | It depends on when spare parts are available |
| EEE repair costs (in €) | 20–150 and sometimes more, Vienna provincial government supports with a sum of 100 maximum the repair costs | 3–50 and sometimes more. No official financial support (informal activity in precarious conditions in Dla and Yde) |
| Availability of EEE spare parts | Yes (Ebay, Amazon, www.ifixit.com, etc.) | Yes, but it takes long time until reception of spare parts, with consequences on the repair time |
| EEE spare parts market | National and international | International |
| EEE spare parts warranty time (in year) | 2–3 (a national law) | Non existant/applicable |
| Labor cost per hour (in €) | 15 without overhead by DRZ and more by some SMEs | 0.3–0.6 (informal activity) |

In view of the above, we conclude that in Austria, plastic, electrical, and electronic waste is already sorted in households, collected, and used to give a second life to either the products or the materials contained in the product. It is also noteworthy that great economic activity occurs in this sector (formal). On the other hand, for a country like Cameroon, like many African countries, very little has been done in this domain, which is still informal. The accumulation of e-waste will become a serious environmental and public health problem in the medium and long term. The questionnaire used in Cameroon shows that there is a huge gap in waste treatment between many African countries and developed countries (e.g., Austria). Africa needs to find measures that can incentivize people to bring their obsolete devices back to collection points and maximize collection. However, most activities in e-waste management in Africa are still informal and thus dangerous for the environment and humans. The value of raw materials presented in electrical and electronic waste worldwide was estimated at US$10 Billion in 2019 [26]. This means that Africa needs to realize that urban mining is not only a way of managing their mining resources in a sustainable manner (resource preservation) but also a way of considerably increasing income when it is done according to the state of the art. In addition, the improvement of repair services in the formal sector through adequate formation can be a source of wealth creation (for example, there were 340 SMEs in the repair business in Austria in 2016, which employed around 1259 people and had an estimated turnover of €113,494,000 [69]). Oluyinka et al. [70] suggests that people who intend to prevent litter are also more likely to factually engage with litter prevention (TPB "Theory of Planned Behavior"), and also that Perceived Behavioral Control (PBC) seems to have a significant impact on the intention to avoid littering. The goal of studies using this theory is to help waste managers formulate policies and interventions that target perceived behavioral intentions in the promotion of waste prevention. Oluyinka et al.'s study demonstrates two things: first, that the intention to prevent waste plays a key role in waste prevention behavior as indicated by TPB, and second, that potential interventions should primarily target people's perception of behavioral control over litter. In addition, environmental managers, applied social and environmental psychologists, and/or social scientists should be involved in designing behavior change programs. According to M. Park et al. [71], e-waste recycling is shifting from the industrialized to the low-cost base of

the developing world, where e-waste recycling if often undertaken in hazardous conditions by a growing informal sector (developing countries offer lower labor costs but inconsistent regulatory enforcement). In the opinion of the authors, designers need to understand how their products end up in waste flows to the developing world and design accordingly for end-of-life. This not only entails the elimination of primary toxic substances within products (as mandated through emerging e-waste regulatory initiatives) but also design for disassembly strategies to eliminate the need for toxic processing and emissions to liberate the valuable recyclates. It would also be interesting to conduct in-depth, concrete, and more representative studies on the African continent over generations to highlight the e-waste management stance.

## 8. Conclusions

African populations, for the most part, live slightly below the poverty line and, consciously or unconsciously, contribute to the protection of the environment simply by preserving their products for as long as possible due to low incomes that prevent them from regularly buying new products. Accustomed to biodegradable organic waste, the African population is not aware of the threat that electronic and electrical waste poses. This means that increased awareness is needed using all necessary means, including the education system, media, social media, workshops, and door-to-door and face-to-face information, to convince the population to behave differently with regard to e-waste. Improved governance developed with industries that consider environmental protection will create many so-called green jobs, with healthy, well-compensated workers, reduce marine pollution, and save species that could otherwise disappear in the long term. Energy savings achieved using recycled materials instead of raw materials are enormous, as shown in Table 3. This is very beneficial to the environment because it allows us to avoid the destruction of landscapes and to reduce $CO_2$ emissions, which are mainly responsible for the climate change that the planet is currently experiencing. Worldwide, eco-design using some DfEs should be implemented in every sector without exception to ensure that the sustainable management of resources and a "circular economy" are achieved.

As a recommendation, strict worldwide regulation and reorganization of e-waste management under the supervision of an international agency, such as the United Nations Environment Programme (UNEP), should be implemented through international legislation. Their mission should be to ensure that in all regions of the globe (or as many as possible) obsolete electronic devices have been collected and properly documented, control their transboundary movement, and produce a yearly report. This ensures that data on this matter are available and accessible to everyone, improves transparency, and raises worldwide awareness of the dangers of e-waste. UNEP should be a real mentor with respect to country/region-based e-waste collection/management at three levels (municipalities, cities, entire countries), advising them on how to use incentives to shift e-waste processing from an informal setting to a formal one as a source of income without endangering the health of workers. At the end of this process, the exact annual quantity of e-scrap produced worldwide should be known, and the next steps in the processing chain can be rationally and sustainably planned.

**Author Contributions:** Conceptualization, G.M.M. and V.-M.A.; Methodology, G.M.M. and V.-M.A.; Software, V.-M.A.; validation, G.M.M. and V.-M.A.; Formal analysis, G.M.M. and V.-M.A.; Investigation, G.M.M.; Resources, G.M.M. and V.-M.A.; Data curation, G.M.M. and V.-M.A.; Writing—original draft preparation, G.M.M.; Writing—review and editing, G.M.M. and V.-M.A.; Visualization, V.-M.A.; supervision, V.-M.A.; Project administration, G.M.M. and V.-M.A. All authors have read and agreed to the published version of the manuscript.

**Funding:** This research received no external funding.

**Data Availability Statement:** Not applicable.

**Acknowledgments:** The authors acknowledge TU Wien University Library for financial support through its Open Access Funding Programme.

**Conflicts of Interest:** The authors declare no conflict of interest.

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
