# Peer review of "Electrical and Electronic Waste Management Problems in Africa: Deficits and Solution Approach"

_environments, doi:10.3390/environments10030044_

Round 1

Reviewer 1 Report

Dear Authors

the work is interesting however serious restructurizing changes and additional literature should be revised as could not waste leading waste management scientists publishing in important journals, e.g., Burlakovs et al., Hogland et al., Kriipsalu et al. Seemingly such works should be referenced to have international scope and avoid geographical referencing gaps

the aim should be specified

more figures and illustrative materials should be added

Author Response

Dear Reviewer

Thank you to the Reviewer for their valuable comments and suggestions. We have added 15 references and upon reconsideration removed 3 based upon the Reviewer's request for additional references. The manuscript has also been checked and rephrased sentence by sentence to improve the English expression. We actively considered the Reviewer's request for restructuring of the manuscript. The nature of the content, however, does not lend itself to a single obvious and superior structure. The interlinkage and co-dependence of the themes covered means that there will always be some degree of compromise in terms of the manuscript structure. We did try multiple different configurations and found that presented in the manuscript to work best. If the Reviewer had specific suggestions on restructuring we would be more than willing to consider and attempt to implement them. Otherwise, we would elect to retain the current manuscript structure. Despite comprehensive examination, we were also unable to find opportunities to add additional figures and illustrations. It should be noted that the manuscript already contains 4 figures and 6 tables and that additional figures and tables would make the manuscript very long and not necessarily relate well to the text, which is already well supported by figures and tables. We are nonetheless willing to consider and attempt to implement requests for specific figures from the Reviewer if they deem illustration of critical points to be lacking and can specify what they would like to see added.

Best regards

Reviewer 2 Report

Review comments on the manuscript: environments-2195343

The paper “ELECTRICAL AND ELECTRONIC WASTE MANAGEMENT PROBLEMS IN AFRICA: DEFICITS AND SOLUTION APPROACH “ is a topic of interest to the readers of Journal Environments, being within the scope of the journal. It is a good and representative review. Needs English revision.

Comments:

. being a review paper, there are several sentences lacking a reference. Just one example: “African continent is with its 53 countries one of the largest (30.37 million km2) and most populous (1.3 billion inhabitants estimated in 2021) continent.”, is this a authors finding? There are several like this.

. do not use “Table 2 will present”, instead “Table 2 present”, once it is on the manuscript; there are other types of similar sentences that need revision

. avoid repetitions, e.g., line 217 is repeated in line 244 (In 2021 the Africa population was estimated of 1.3 billion inhabitants). A considerable text review must be done

. Line 408 – do not use “ In view of the above,” when referring to a table, instead use its number (e.g., line 403). Restructure the sentences.

Author Response

Thank you to the Reviewer for their valuable comments and suggestions. The English expression in the manuscript has been extensively revised, sentence by sentence, at the Reviewer's request. This also includes the specified changes requested by the Reviewer. We have also added 15 references and upon reconsideration removed 3 based upon the Reviewer's request for additional references.

Reviewer 3 Report

The manuscript will benefit from mild editing by a native English speaker or professional editor/translator.

Author Response

Thank you to the Reviewer for their valuable comments and suggestions. We are grateful for their time in reviewing our manuscript and their appreciation of its value. The manuscript has also been checked and rephrased sentence by sentence to improve the English expression.

Reviewer 4 Report

This review is interesting and well-presented. This refers to the electronic and electrical waste generated and/or transported in Africa, waste treatment and recycling with methods for recovering metals from e-waste. The roles and responsibilities of stakeholders and institutions involved, especially for Africa, are discussed. 

I propose the acceptance of this review in its present form.

Author Response

Thank you to the Reviewer for their overt praise of our manuscript. We are especially grateful for their time in reviewing our manuscript and their appreciation of its value.

Round 2

Reviewer 1 Report

Good Job!

Think is good to be published

Reviewer 2 Report

Thank for the reviewed version. Document improved notably. Good work!